# Flexible and Efficient Inference with Particles for the Variational Gaussian Approximation

**DOI:** 10.3390/e23080990

**Published:** 2021-07-30

**Authors:** Théo Galy-Fajou, Valerio Perrone, Manfred Opper

**Affiliations:** 1Artificial Intelligence Group, Technische Universität Berlin, 10623 Berlin, Germany; manfred.opper@tu-berlin.de; 2Amazon Web Services, 10969 Berlin, Germany; vperrone@amazon.com; 3Centre for Systems Modelling and Quantitative Biomedicine, University of Birmingham, Birmingham B15 2TT, UK

**Keywords:** variational inference, Gaussian, particle flow, variable flow

## Abstract

Variational inference is a powerful framework, used to approximate intractable posteriors through variational distributions. The de facto standard is to rely on Gaussian variational families, which come with numerous advantages: they are easy to sample from, simple to parametrize, and many expectations are known in closed-form or readily computed by quadrature. In this paper, we view the Gaussian variational approximation problem through the lens of gradient flows. We introduce a flexible and efficient algorithm based on a linear flow leading to a particle-based approximation. We prove that, with a sufficient number of particles, our algorithm converges linearly to the exact solution for Gaussian targets, and a low-rank approximation otherwise. In addition to the theoretical analysis, we show, on a set of synthetic and real-world high-dimensional problems, that our algorithm outperforms existing methods with Gaussian targets while performing on a par with non-Gaussian targets.

## 1. Introduction

Representing uncertainty is a ubiquitous problem in machine learning. Reliable uncertainties are key for decision making, especially in contexts where the trade-off between exploitation and exploration plays a central role, such as Bayesian optimization [1], active learning [2], and reinforcement learning [3]. While Bayesian inference is a principled tool to provide uncertainty estimation, computing posterior distributions is intractable for many problems of interest. Most sampling methods struggle to scale up to large datasets [4], while the diagnosis of convergence is not always straightforward [5]. On the other hand, Variational Inference **(VI)** methods can rely on well-understood optimization techniques and scale well to large datasets, at the cost of an approximation quality depending heavily on the assumptions made. The Gaussian family is by far the most popular variational approximation used in VI [6,7]. This is for several reasons. First, Gaussian variational families are easy to sample from, reparametrize, and marginalize. Second, they are easily amenable to diagonal covariance approximations, making them scalable to high dimensions. Third, most expectations are either easily computable by quadrature or Monte Carlo integration, or known in closed-form.

A large body of work covers different approaches to optimize the Variational Gaussian Approximation ************(VGA)************, with the speed of convergence and the scalability in dimensions as the main concerns. From the perspective of convergence speed, the major bottleneck when computing gradients with stochastic estimators is the estimator variance [8]. *Particle-based methods* with deterministic paths do not have this issue, and have been proven to be highly successful in many applications [9,10,11]. However, can we use a particle-based algorithm to compute a VGA? If so, what are its properties and is it competitive with other VGA methods?

In this paper, we attempt to answer these questions by introducing the Gaussian Particle Flow ****(GPF)****, a framework to approximate a Gaussian variational distribution with particles. GPF is derived from a continuous-time flow, where the necessary expectations over the evolving densities are approximated by particles. The complexity of the method grows quadratically with the number of particles but linearly with the dimension, remaining compatible with other approximations such as structured mean-field approximations. Using the same dynamics, we also derive a stochastic version of the algorithm, Gaussian Flow ****(GF)****. To show convergence, we prove the decrease in an empirical version of the free energy that is valid for a finite number of particles. For the special case of *D*–dimensional Gaussian target densities, we show that D+1 particles are enough to obtain convergence to the true distribution. We also find, for this case, that convergence is exponentially fast. Finally, we compare our approach with other VGA algorithms, both in fully controlled synthetic settings and on a set of real-world problems.

## 2. Related Work

The goal of Bayesian inference is to carry out computations with the posterior distribution of a latent variable $x\in R^{D}$ given some observations *y*. By Bayes theorem, the posterior distribution is $p\left(x|y\right)=\frac{p(y|x)p(x)}{p(y)}$, where p(y|x) and p(x) are, respectively, the likelihood and the prior distribution. Even if the likelihood and the prior are known analytically, marginalizing out high-dimensional variables in the product p(y|x)p(x) in order to compute quantities such as p(y) is typically intractable. Variational Inference **(VI)** aims to simplify this problem by turning it into an optimization one. The intractable posterior is approximated by the closest distribution within a tractable family, with closeness being measured by the Kullback-Leibler **(KL)** divergence, defined by
$$\begin{array}{c} \mathrm{KL}\;\left[q(x)||p(x)\right]=E_{q}\left[\log q(x)-\log p(x)\right], \end{array}$$
where $E_{q}\left[f(x)\right]=\int f\left(x\right)q\left(x\right)dx$ denotes the expectation of *f* over *q*. Denoting by Q a family of distributions, we look for
$$\begin{array}{c} \underset{q\in Q}{\arg\min}\;\mathrm{KL}\;\left[q(x)||p(x|y)\right]. \end{array}$$

Since p(y) is not computable in an efficient way, we equivalently minimize the upper bound F:(1)$$\begin{array}{c} \mathrm{KL}\left[q(x)||p(x|y)\right]\leq F\left[q\right]=-E_{q}\left[\log p(y|x)p(x)\right]-H_{q}, \end{array}$$
where $H_{q}$ is the entropy of *q* ($-E_{q}\left[\log q(x)\right]$). Here, F is known as the variational free energy and −F is known as the Evidence Lower BOund (ELBO). A diverse set of approaches to perform VI with Gaussian families Q have been developed in the literature, which we review in the following.

### 2.1. The Variational Gaussian Approximation

The VGA is the restriction of Q to be the family of multivariate Gaussian distributions q(x)=N(m,C), where $m\in R^{D}$ is the mean and $C\in \{A\in R^{D\times D}|x^{⊤}Ax\geq 0,\forall x\in R^{D}\}$ is the covariance matrix, for which the free energy is found to be
(2)$$\begin{array}{c} F\left[q\right]=-\frac{1}{2}\log\left|C\right|+E_{q}\left[\varphi (x)\right]. \end{array}$$
where φ(x)=−log(p(y|x)p(x)). A standard descent algorithm based on gradients of Equation (Equation 2) with respect to variational parameters m,C give rise to some issues. First, naively computing the gradient of the expectation with respect to the covariance matrix *C* involves unwanted second derivatives of φ(x) [12], which may not be available or may be computationally too expensive in a *black-box* setting. Second, the gradient of the entropy term $H_{q}$ entails inverting a non-sparse matrix, which we would like to avoid for higher-dimensional cases. Finally, the positive-definiteness of the covariance matrix leads to non-trivial constraints on parameter updates, which can lead to a slowdown of convergence or, if ignored, to instabilities in the algorithm.

To solve these issues, a variety of approaches have been proposed in the literature. If we focus on factorizable models, we can make a simplification: for problems with likelihoods that can be rewritten as $p\left(y|x\right)=\prod_{d=1}^{D}p\left(y|x_{d}\right)$, the number of independent variational parameters is reduced to 2D [12,13]. In this special case, the Gaussian expectations in the free energy (Equation 2) split into a sum of 1-dimensional integrals, which can be efficiently computed by using numerical quadrature methods. To extend to the general case, gradients of the free energy are estimated by a stochastic sampling approach, which also forms the starting point of our method. This relies on the so-called *reparametrization trick*, where the expectation over the parameter-dependent variational density $q_{\theta }$ is replaced by an expectation over a fixed density $q^{0}$ instead. This facilitates the gradient computation because unwanted derivatives of the type $\nabla_{\theta }q_{\theta }\left(x\right)$ are avoided. For the Gaussian case, the reparametrization trick is a linear transformation of an arbitrary *D* dimensional Gaussian random variable $x∼q_{\theta }\left(x\right)$ in terms of a *D*-dimensional Gaussian random variable $x^{0}∼q^{0}=N\left(m^{0},C^{0}\right)$:(3)$$\begin{array}{c} x=\Gamma (x^{0}-m^{0})+m, \end{array}$$
where $\Gamma \in R^{D\times D}$ and $m\in R^{D}$ are the variational parameters. We assume that the covariance $C^{0}$ is not degenerate and, for simplicity, we set it as the identity. For instance, the gradient of the expectation given *q* over a function *f* given the mean *m* becomes $\nabla_{m}E_{q}\left[f(x)\right]=E_{q^{0}}\left[\nabla_{m}f\left(\Gamma \left(x^{0}-m^{0}\right)+m\right)\right]$. This can be simply proved by using the reparametrization (Equation 3) inside the integral and passing the gradient inside; for more details, see [14].

Given this representation, the free energy is easily obtained as a function of the variational parameters:(4)$$\begin{array}{c} F\left(q\right)=-\log|\Gamma |+E_{q^{0}}\left[\varphi (\Gamma \left(x^{0}-m^{0}\right)+m)\right]. \end{array}$$Other representations are possible. Challis and Barber [13] and Ong et al. [15] use a different reparametrization with a factorized structure of the covariance $C=\Gamma^{⊤}\Gamma +\mathrm{diag}\left(d\right)$, where $\Gamma \in R^{D\times P}$ and $d\in R^{D}$, with P≤D is the rank of $\Gamma^{⊤}\Gamma$. Other representations assume special structures of the precision matrix $\Lambda =C^{-1}$, which allow you to enforce special properties, such as sparsity in [16,17].

In general, these methods tend to scale poorly with the number of dimensions, as one needs to optimize D(D+3)/2 parameters. The (structured) Mean-Field **(MF)** [18,19] approach imposes independence between variables in the variational distribution. The number of variational parameters is then 2D, but covariance information between dimensions is lost.

### 2.2. Natural Gradients

Besides the issue of expectations, more efficient optimizations directions, beyond ordinary gradient descent, have been considered. These can help to deal with constraints such as those given for the covariance matrix. Natural gradients [20] are a special case of Riemannian gradients and utilize the specific Riemannian manifold structure of variational parameters. They can often deal with constraints of parameters (such as the positive definiteness of the covariance), accelerate inference, and improve the convergence of algorithms. The application of such advanced gradient methods typically requires an estimate of the inverse Fisher information matrix as a preconditioner of ordinary gradients. Khan and Nielsen [21] and Lin et al. [22] propose a solution that requires extra second derivatives of the log–posteriors. Salimbeni et al. [23] developed an automatic process to compute these without the second derivatives but with instability issues. Lin et al. [17] solved these issues by using geodesics on the manifold of parameters, at the price of having to compute inverse matrices as well as Hessians.

### 2.3. Particle-Based VI

Stochastic gradient descent methods compute expectations (and gradients) at each time step with new independent Monte Carlo samples drawn from the current approximation of the variational density. Particle-based methods for variational inference *draw samples only once* at the beginning of the algorithm instead. They iteratively construct transformations of an initial random variable (having a simple tractable density) where the transformed density leads to the decrease and finally to the minimum of the variational free energy. The iterative approach induces a deterministic temporal flow of random variables which depends on the current density of the variable itself. Using an approximation by the empirical density (which is represented by the positions of a set of ’particles’) one obtains a flow of interacting particles which converges asymptotically to an empirical approximation of the desired optimal variational density.

The most popular approach is Stein Variational Gradient Descent ******(SVGD)****** [24], which computes a nonparametric transformation based on the kernelized Stein discrepancy [9]. SVGD has the advantage of not being restricted to a parametric form of the variational distribution. However, using standard distance-based kernels like the squared exponential kernel ($k\left(x,y\right)=\exp(-∥x-y{∥}_{2}^{2}/2)$) can lead to underestimated covariances and poor performance in high dimensions [11,25]. Hence, it is interesting to develop particle approaches that approximate the VGA. We provide a more thorough comparison between our method and SVGD in Section 3.6.

### 2.4. GVA in Bayesian Neural Networks

There has been increased interest in making Bayesian Neural Networks **(BNN)** by adding priors to Neural Networks parameters. The true form of the posterior is unknown but VGA has been used due to its ease of use and scalability with the number of dimensions (typically $D\gg 10^{5}$). Most of the aforementioned methods apply to BNN, but techniques have been specifically tailored with BNN in mind. [26] use the low-rank structure of [13] but exploit the *Local Reparametrization Trick*, where each datapoint $y_{i}$ gets a different sample from *q* in order to reduce the stochastic gradient estimator variance. *Stochastic Weight Averaging-Gaussian*
**(SWAG)** [27], in which a set of particles obtained via stochastic gradient descent represent a low-rank Gaussian distribution, approximating the true posterior with a prior posterior produced by the network’s regularization. While easy to implement, SWAG does not allow you to incorporate an explicit prior, and the resulting distribution does not derive from a principled Bayesian approach.

### 2.5. Related Approaches

The closest approach to our proposed method is the *Ensemble Kalman Filter*
**(EKF)** [28]. It assumes that the posterior is computed in a sequential way, where, at each time step, only single (or smaller batches) of data observations, represented by their likelihoods, become available. An ensemble of particles, representing a Gaussian distribution is iteratively updated with every new batch of observations. EKF allows us to work on high-dimensional problems with a limited amount of particles but is restricted to factorizable likelihoods for which a sequential representation is possible. While EKF maintains a representation of a Gaussian posterior, it is not clear how this relates to the goal of minimizing the free energy or the KL divergence.

## 3. Gaussian (Particle) Flow

We introduce *Gaussian Particle Flow*
**(GPF)** and *Gaussian Flow*
**(GF)**, two computationally tractable approaches, to obtain a *Variational Gaussian Approximation ***(VGA)**. In the following, we derive deterministic linear dynamics, which decreases the variational free energy. We additionally give some variants with a *Mean-Field *
**(MF)** approach and prove theoretical convergence guarantees.

In the following, $\frac{d(\cdot )}{dt}$ indicates the total derivative given time, $\frac{\partial (\cdot )}{\partial t}$ partial derivatives given time, $\nabla_{x}\left(\cdot \right)$ gradients given a vector *x*.

### 3.1. Gaussian Variable Flows

We next discuss an alternative approach to generate the desired transformation of random variables, leading from a simple (prior) Gaussian density to a more complex Gaussian, which minimizes the variational free energy. It is based on the idea of *variable flows*, i.e., recursive deterministic transformations of the random variables defined by a mapping $x^{n+1}=x^{n}+\epsilon f^{n}\left(x^{n}\right)$ where $f^{n}:R^{D}\to R^{D}$. Well-known examples of flows are *Normalizing Flows* [29], where $f^{n}$ are bijections, or *Neural ODEs* [30] where $f^{n}=f$ is defined by a neural network and $x^{0}$ is the input. For simplicity, we will consider small changes ϵ→0 and work with flows in the continuous-time limit (t=nϵ), which follow a system of *Ordinary Differential Equation*
**(ODE)**. For the Gaussian case, in the spirit of the reparametrization trick (Equation 3), we choose a linear corresponding map *f* and write
(5)$$\begin{array}{c} \frac{dx^{t}}{dt}=f^{t}\left(x^{t}\right)=A^{t}\left(x^{t}-m^{t}\right)+b^{t}, \end{array}$$
where $A^{t}$ is a matrix and $m^{t}≐E_{q^{t}}\left[x\right]$ (which is no longer interpreted as an independent variational parameter). When the initial random variable $x^{0}$ is Gaussian distributed, the vectors $x^{t}$ are also Gaussian for any *t*. To construct a flow that decreases the free energy over time, we can either compute the time derivative of the specific free energy (Equation 2) induced by the ODE (Equation 5), or simply derive the general result valid for smooth maps *f* (see, e.g., [24]). To be self contained, we briefly repeat the main steps: We first compute the change of the free energy in terms of the time derivative of $q^{t}$:$$\begin{array}{cc} \frac{dF[q^{t}]}{dt}= & \frac{d}{dt}\int q^{t}\left(x\right)\left(\log q^{t}\left(x\right)+\varphi \left(x\right)\right)dx\\ = & \int \frac{\partial q^{t}\left(x\right)}{\partial t}\left(\log q^{t}\left(x\right)+\varphi \left(x\right)\right)dx+\int q^{t}\left(x\right)\left(\frac{\partial q^{t}\left(x\right)}{\partial t}\frac{1}{q^{t}\left(x\right)}+\frac{\partial \varphi (x)}{\partial t}\right)dx\\ = & \int \frac{\partial q^{t}\left(x\right)}{\partial t}\left(\log q^{t}\left(x\right)+\varphi \left(x\right)\right)dx \end{array}$$
where we have used the fact that $\int \frac{\partial q^{t}\left(x\right)}{\partial t}dx=\frac{d}{dt}\int q^{t}\left(x\right)dx=0$ and $\frac{\partial \varphi (x)}{\partial t}=0$. We next use the *continuity equation* for the density
$$\begin{array}{c} \frac{\partial q^{t}\left(x\right)}{\partial t}=-\nabla_{x}\cdot \left(q^{t}\left(x\right)f^{t}\left(x\right)\right), \end{array}$$
related to the deterministic flow to obtain
$$\begin{array}{cc} \frac{dF[q^{t}]}{dt}= & \int \nabla_{x}\cdot \left(q^{t}\left(x\right)f^{t}\left(x\right)\right)\left(\log q^{t}\left(x\right)+\varphi \left(x\right)\right)dx\\ = & -\int \left(q^{t}\left(x\right)f^{t}\left(x\right)\right)\cdot \nabla_{x}\left(\log q^{t}\left(x\right)+\varphi \left(x\right)\right)dx\\ = & \int \left(\nabla_{x}\cdot \left(q^{t}\left(x\right)f^{t}\left(x\right)\right)+q^{t}\left(x\right)f^{t}\left(x\right)\cdot \nabla_{x}\varphi \left(x\right)\right)dx\\ = & \int \nabla_{x}q^{t}\left(x\right)\cdot f^{t}\left(x\right)+q^{t}\left(x\right)f^{t}\left(x\right)\cdot \nabla_{x}\varphi \left(x\right)dx\\ = & -E_{q^{t}}\left[\nabla_{x}\cdot f^{t}\left(x\right)-f^{t}\left(x\right)\cdot \nabla_{x}\varphi \left(x\right)\right] \end{array}$$
where we have applied Green’s identity twice and used the fact that $\lim_{x\to \infty }q_{t}\left(x\right)=0$. Specializing to the linear flow (Equation 5), we obtain   
(6)$$\begin{array}{c} \frac{dF[q^{t}]}{dt}=-\mathrm{tr}\left[A^{t}{\left(A_{★}^{t}\right)}^{⊤}\right]-{\left(b^{t}\right)}^{⊤}b_{★}^{t}, \end{array}$$
where
(7)$$\begin{array}{cc} A_{★}^{t}≐ & I-E_{q^{t}}\left[\nabla_{x}\varphi \left(x\right){\left(x-m^{t}\right)}^{⊤}\right]\\ b_{★}^{t}≐ & -E_{q^{t}}\left[\nabla_{x}\varphi \left(x\right)\right] \end{array}$$Equation (Equation 6) represents the change in the free energy F for an infinitesimal change in the variables *x* given by the flow (Equation 5). Obviously, the simplest choices
(8)$$\begin{array}{c} A^{t}\equiv A_{★}^{t}\;b^{t}\equiv b_{★}^{t} \end{array}$$
lead to a decrease in the free energy $\frac{dF[q^{t}]}{dt}\leq 0$. More detailed derivations are given in Appendix A. Additionally, *equality* only happens, when
(9)$$\begin{array}{c} I-E_{q}\left[\nabla_{x}\varphi \left(x\right){\left(x-m\right)}^{⊤}\right]=0\\ E_{q}\left[\nabla_{x}\varphi \left(x\right)\right]=0 \end{array}$$Using Stein’s lemma [31], we can show that these fixed-point solutions are equal to the conditions for the optimal variational Gaussian distribution solution given in [12]. In Appendix C, we show that our parameter updates can be interpreted as a Riemannian gradient descent method for the free energy (Equation 4). This is based on the metric introduced by ([20], Theorem 7.6) as an efficient technique for learning the mixing matrix in models of blind source separation. This gradient should not be confused with the so-called *natural gradient* obtained by pre-multiplying with the inverse Fischer-information matrix.

Of course, there are other choices for $A^{t}$ and $b^{t}$, which lead to a decrease in the free energy and the same fixed-point equations. In Section 3.6, we discuss how SVGD, with a linear kernel, can lead to the same fixed points but with different dynamics.

### 3.2. From Variable Flows to Parameter Flows

Before we introduce the particle algorithm, we show that the results for the variable flow can also be converted into a temporal change of the parameters $\Gamma^{t}$, $m^{t}$, as defined for Equation (Equation 3). From this, a corresponding Gaussian Flow ****(GF)**** algorithm can be easily derived. By differentiating the parametrisation $x^{t}=\Gamma^{t}\left(x^{0}-m^{0}\right)+m^{t}$ (with $m^{t}$ now considered as free variational parameter) with respect to time *t* and using (Equation 5), we obtain
(10)$$\begin{array}{c} \frac{dx^{t}}{dt}=\frac{d\Gamma^{t}}{dt}\left(x^{0}-m^{0}\right)+\frac{dm^{t}}{dt}=A^{t}\left(x^{t}-m^{t}\right)+b^{t} \end{array}$$By inserting $x^{t}=\Gamma^{t}\left(x^{0}-m^{0}\right)+m^{t}$ into the right hand side of (Equation 10), and using the optimal parameters from (Equation 7), we obtain
(11)$$\begin{array}{c} \begin{array}{cc} \frac{d\Gamma^{t}}{dt}= & \Gamma^{t}-E_{q^{0}}\left[\nabla_{x}\varphi \left(x^{t}\right){\left(x^{0}-m^{0}\right)}^{⊤}\right]\Gamma^{t}{\left(\Gamma^{t}\right)}^{⊤}\\ \frac{dm^{t}}{dt}= & -E_{q^{0}}\left[\nabla_{x}\varphi \left(x^{t}\right)\right] \end{array} \end{array}$$Note that the expectations are over the probability distribution of the initial random variable $x^{0}$. Discretizing Equations (Equation 11) in time, and estimating the expectations by drawing independent samples from the fixed Gaussian $q^{0}$ at each time step, we obtain our GF algorithm to minimize the variational free energy in the space of Gaussian densities. We summarize the steps of GF in Algorithm 1. Remarkably, this scheme differs from previous VGA algorithms with Riemannian gradients based on the Fisher information metric (see, e.g., [17,32]) because no *matrix inversions* or *second order derivatives* of the function φ are required.

GF also allows for the computation of a low-rank VGA by enforcing $\Gamma \in R^{D\times K}$ and $x^{0}\in R^{K}$. This algorithm scales linearly in the number of dimensions and quadratically in the rank *K* of the covariance.

It is interesting to note that the reverse construction of a variable flow from a parameter flow is, in general, not possible. This would require the ability to eliminate all variational parameters and the initial variables $x^{0}$ in the resulting differential equation for $x^{t}$, and replace them with functions of $x^{t}$ alone. For instance, if we eliminate the initial variables $x^{0}$ in terms of ${\left(\Gamma^{t}\right)}^{-1}$ and $x^{t}$ the algorithm of [14], the resulting expression still depends on $\Gamma^{t}$.

### 3.3. Particle Dynamics

The main idea of the particle approach is to approximate the Gaussian density $q^{t}$ in (Equation 7) by the empirical distribution
(12)$$\begin{array}{c} {\hat{q}}^{t}≐\frac{1}{N}\sum_{i=1}^{N}\delta \left(x-x_{i}^{t}\right) \end{array}$$
computed from *N* samples $x_{i}^{t}$, i=1,…,N. These are initially sampled from the density $q^{0}$ at time t=0 and are then propagated using the discretized dynamics of the ODE (Equation 5):(13)$$\begin{array}{c} \frac{dx_{i}^{t}}{dt}=-\eta_{1}^{t}E_{{\hat{q}}^{t}}\left[\nabla_{x}\varphi \left(x\right)\right]-\eta_{2}^{t}{\hat{A}}^{t}\left(x_{i}^{t}-{\hat{m}}^{t}\right) \end{array}$$
where
$$\begin{array}{c} {\hat{A}}^{t}=I-\frac{1}{N}\sum_{i=1}^{N}\nabla_{x}\varphi \left(x\right){\left(x_{i}^{t}-{\hat{m}}^{t}\right)}^{⊤}\\ {\hat{b}}^{t}=\frac{1}{N}\sum_{i=1}^{N}\nabla_{x}\varphi \left(x_{i}^{t}\right),\;{\hat{m}}^{t}=\frac{1}{N}\sum_{i=1}^{N}x_{i}^{t} \end{array}$$
where $\eta_{1}^{t}$ and $\eta_{2}^{t}$ are learning rates (We further comment on the use of different optimization schemes in Section 4.4). Note that although $E_{{\hat{q}}^{t}}\left[\nabla_{x}\varphi \left(x\right){\left(x-{\hat{m}}^{t}\right)}^{⊤}\right]$ is a D×D matrix, changing the matrix multiplication order leads to a computational complexity of $O(N^{2}D)$ with a storage complexity of O(N(N+D)), since neither the empirical covariance matrix or $A^{t}$ need to be explicitly computed.

#### Relaxation of Empirical Free Energy and Convergence

We have shown that the continuous-time dynamics (Equation 10) of the random variables leads to a decay of the free energy $F(q^{t})$ with time *t*. Assuming that the free energy is bounded from below, one might conjecture that this property would imply the convergence of the particle algorithm to a fixed point when learning rates are sufficiently small such that the discrete-time dynamics are approximated well by the continuous limit. Unfortunately, the finite number *N* of particles poses an extra problem. The definition of the free energy F(q) by the KL–divergence (Equation 1) for continuous random variables such as assumes that both q(·) and p(·|y) are densities with respect to the Lebesgue measure. Hence, $F(\hat{q})$ is not defined if we take $q\equiv \hat{q}$, (Equation 12) as the empirical distribution of the finite particle approximation. Nevertheless, we define a finite *N* *approximation* to the Gaussian free energy, which is also then found to decay under the finite *N* dynamics. Let us first assume that N>D and define
(14)$$\begin{array}{c} \tilde{F}\left({\hat{q}}^{t}\right)≐-\frac{1}{2}\log\left|{\hat{C}}^{t}\right|+E_{{\hat{q}}^{t}}\left[\varphi (x)\right] \end{array}$$
with the empirical covariance matrix
(15)$$\begin{array}{c} {\hat{C}}^{t}=\frac{1}{N}\sum_{i=1}^{N}\left(x_{i}^{t}-m^{t}\right){\left(x_{i}^{t}-m^{t}\right)}^{⊤} \end{array}$$The definition (Equation 14) is chosen in such way that in the large *N* limit, when the empirical distribution ${\hat{q}}^{t}$ converges to a Gaussian distribution $q^{t}$, we will also obtain the convergence of the approximation (Equation 14) to $F(q^{t})$. It can be shown (see Appendix B) that $\frac{d\tilde{F}\left({\hat{q}}^{t}\right)}{dt}\leq 0$, with equality only at the fixed points of the dynamics.

In applications of our particle method to high-dimensional problems, the limitations of computational power may force us to restrict particle numbers to be smaller than the dimensionality *D*. For N<D+1, the empirical covariance $C^{t}$ will be singular, and typically contain only N−1 non-zero eigenvalues, which leads to the $-\log\left|\hat{C}\right|=\infty$ and makes Equation (Equation 14) meaningless. We resolve this issue through a regularisation of the log–determinant term in (Equation 14), replacing all zero eigenvalues of $\hat{C}$ by the values 1, i.e., $\lambda_{i}=0\to {\tilde{\lambda }}_{i}=1$. We show in Appendix B that the free energy still decays, provided that the dynamics of the particles stay the same. This regularisation step can be formally stated as a replacement of the empirical covariance (Equation 15) in (Equation 14) by
$$\begin{array}{c} \hat{C^{t}}\to {\hat{C}}^{t}+\sum_{i:\lambda_{i}^{t}=0}e_{i}^{t}{\left(e_{i}^{t}\right)}^{⊤} \end{array}$$
where $e_{i}^{t}=$*i*th eigenvector of ${\hat{C}}^{t}$.

### 3.4. Algorithm and Properties

The algorithm we propose is to sample *N* particles $\left\{x_{1}^{0},\ldots ,x_{N}^{0}\right\}$ where $x_{i}^{0}\in R^{D}$ from $q^{0}$ (which can be centered around the MAP for example), and iteratively optimize their positions using Equation (Equation 13). Once convergence is reached, i.e., $\frac{dF}{dt}=0$, we can easily make predictions using the converged empirical distribution $\hat{q}\left(x\right)=\frac{1}{N}\sum_{i=1}^{N}\delta \left(x-x_{i}\right)$, where δ is the Dirac delta function, or, alternatively, the Gaussian density it represents, i.e., q(x)=N(m,C), where $m=\frac{1}{N}\sum_{i=1}^{N}x_{i}$ and $C=\frac{1}{N}\sum_{i=1}^{N}\left(x_{i}-m\right){\left(x_{i}-m\right)}^{⊤}$. To draw samples from $\hat{q}$, no inversions of the empirical covariance *C* are needed, as we can obtain new samples by computing:(16)$$\begin{array}{c} x=\frac{1}{\sqrt{N}}\sum_{i=1}^{N}\left(x_{i}-m\right)\circ \xi_{i}+m, \end{array}$$
where $\xi_{i}$ are i.i.d. normal variables: $\xi_{i}∼N\left(0,I_{D}\right)$. This can be shown by defining *D*, the deviation matrix, a matrix which columns equal to $D_{i}=\frac{x_{i}-m}{\sqrt{N}}$. We naturally have $DD^{⊤}=C$ which makes *D* the Cholesky decomposition of *C*.

All the inference steps are summarized in Algorithm 2 and an illustration in two dimensions is provided in Figure 1.

We summarize the principal points of our approach:Gradients of expectations have zero variance, at the cost of a bias decreasing with the number of particles and equal to zero for Gaussian target (see Theorem 1);It works with noisy gradients (when using subsampling data, for example);The rank of the approximated covariance *C* is min(N−1,D). When N≤D, the algorithm can be used to obtain a low-rank approximation.The complexity of our algorithm is $O(N^{2}D)$ and storing complexity is O(N(N+D)). By adjusting the number of particles used, we can control the performance trade-off;GPF (and GF) are also compatible with any kind of structured MF (see Section 3.5);Despite working with an empirical distribution, we can compute a surrogate of the free energy F(q) to optimize hyper-parameters, compute the lower bound of the log-evidence, or simply monitor convergence.
**Algorithm 1:** Gaussian Flow **(GF)**
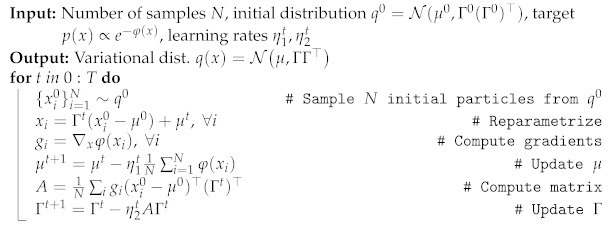


**Algorithm 2:** Gaussian Particle Flow **(GPF)**

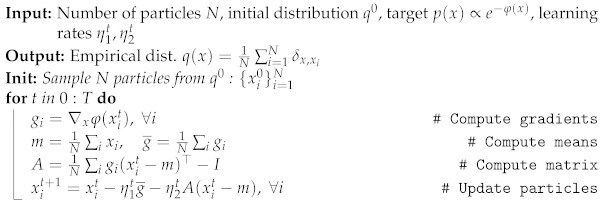



#### 3.4.1. Relaxation of Empirical Free Energy

The definition of the free energy F(q) from the KL–divergence (Equation 1) for a continuous random variables assumes that both q(·) and p(·|y) are densities with respect to the Lebesgue measure. Hence, it is not *a priori* clear that a specific *approximation* $F({\hat{q}}^{t})$, based on an empirical distribution ${\hat{q}}^{t}\left(x\right)≐\frac{1}{N}\sum_{i=1}^{N}\delta \left(x-x_{i}^{t}\right)$ with a *finite number of particles*
*N*, will decrease under the particle flow. Thus we may not be able to guarantee convergence to a fixed point for finite *N*. Luckily, as we show in Appendix D, we find that:   
(17)$$\begin{array}{c} \frac{dF({\hat{q}}_{t})}{dt}=\frac{d(E_{{\hat{q}}_{t}}\left[\varphi (x)\right]-\frac{1}{2}\log\left|C^{t}\right|)}{dt}\leq 0. \end{array}$$ For N<D+1, the empirical covariance $C^{t}$ will typically contain N−1 non-zero eigenvalues and lead to $-\log\left|C\right|=\infty$, making Equation (Equation 17) meaningless. We resolve this issue by introducing a *regularized free energy*
$\tilde{F}$ where $\log\left|C^{t}\right|$ is replaced by $\sum_{i:\lambda_{i}>0}\log\lambda_{i}$ where ${\left\{\lambda_{i}\right\}}_{i=1}^{D}$ are the eigenvalues of $C^{t}$. We show in Appendix D that, given the dynamics from Equation (Equation 5), $\tilde{F}$ is also guaranteed to not increase over time. It can, therefore, be used as a regularized proxy for the true F and used to optimize over hyper-parameters or to monitor convergence. Note that similar proofs exist for SVGD [33] and were proven to be highly non-trivial.

#### 3.4.2. Dynamics and Fixed Points for Gaussian Targets

We illustrate our method by some exact theoretical results for the dynamics and the fixed points of our algorithm when *the target is a multivariate Gaussian density*. While such targets may seem like a trivial application, our analysis could still provide some insight into the performance for more complicated densities.

**Theorem** **1.**
*If the target density p(x) is a D-dimensional multivariate Gaussian, only D+1 particles are needed for Algorithm 2 to converge to the exact target parameters.*


**Proof.** The proof is given in Appendix E.    □

**Theorem** **2.**
*For a target $p\left(x\right)=N\left(x|\mu ,\Lambda^{-1}\right)$, i.e., with precision matrixΛ, where $x\in R^{D}$, and N≥D+1 particles, the continuous time limit of Algorithm 2 will converge exponentially fast for both the mean and the trace of the precision matrix:*
$$\begin{array}{cc} m^{t}-\mu = & e^{-\Lambda t}\left(m^{0}-\mu \right),\\ \mathrm{tr}({\left(C^{t}\right)}^{-1}-\Lambda )= & e^{-2t}\mathrm{tr}\left({\left(C^{0}\right)}^{-1}-\Lambda \right), \end{array}$$
*where $m^{t}$ and $C^{t}$ are the empirical mean and covariance matrix at time t and exp(−Λt) is the matrix exponential.*


**Proof.** The proof is given in Appendix F.    □

Our result shows that convergence of the mean $m^{t}$ directly depends on Λ. However, we can also precondition the gradient on *m* by $C^{t}$, i.e., using the natural gradient approximation in the Fisher sense, and eventually get rid of the dependency on Λ when ${\left(C^{t}\right)}^{-1}\approx \Lambda$.

The exponential relaxation of fluctuations also manifests itself in the decay of the free energy towards its minimum. For the Gaussian target, the free energy exactly separates into two terms corresponding to the mean and fluctuations. We can write $F\left(m^{t},C^{t}\right)=\frac{1}{2}{\left(m^{t}-\mu \right)}^{⊤}\Lambda \left(m^{t}-\mu \right)+\frac{D}{2}+F_{fl}\left(C^{t}\right)$, where the nontrivial fluctuation part (subtracted by its minimum) is given by
$$\begin{array}{c} F_{fl}\left(C^{t}\right)=-\frac{1}{2}\log\left|C^{t}\right|+\frac{1}{2}\mathrm{tr}\left(\Lambda C^{t}-I\right). \end{array}$$We can show that
$$\begin{array}{c} -\lim_{t\to \infty }\frac{d\ln F_{fl}\left(C^{t}\right)}{dt}\geq 4, \end{array}$$
indicating an asymptotic decrease in $F_{fl}\left(C^{t}\right)$ faster than $e^{-4t}$, independent of the target. We can also prove the finite time bound   
$$\begin{array}{c} F_{fl}\left(C^{t}\right)\leq F_{fl}\left(C^{0}\right)e^{-\left[\frac{2t}{\mathrm{tr}\left(\Lambda^{-1}\right)(\mathrm{tr}\left(\Lambda \right)+|\mathrm{tr}\left({\left(C^{0}\right)}^{-1}-\Lambda \right)\left|\right)}\right]}. \end{array}$$The degenerate case N<D+1

Additionally, we can show the following result for the fixed points:

**Theorem** **3.**
*Given a D-dimensional multivariate Gaussian target density p(x)=N(x|μ,Σ), using Algorithm 2 with N<D+1 particles, the empirical mean converges to the exact mean μ. The N−1 non-zero eigenvalues of $C^{t}$ converge to a subset of the target covariance Σ spectrum. Furthermore, the *
***global minimum***
*of the regularised version $\tilde{F}$ of the free energy (Equation 17) corresponds to the *
***largest***
* eigenvalues of Σ.*


**Proof.** The proof is given in Appendix G.    □

This result suggests that $C^{t}$ might typically converge to an optimal low-rank approximation of Σ. We show an empirical confirmation in Section 4.2 for this conjecture. This suggests that it makes sense to apply our algorithm to high-dimensional problems even when the number of particles is not large. If the target density has significant support close to a low-dimensional submanifold, we might still obtain a reasonable approximation.

### 3.5. Structured Mean-Field

For high-dimensional problems, it may be useful to restrict the variational Gaussian approximation to the posterior to a specific structure via a structured mean-field approximation. In this way, spurious dependencies between variables that are caused by finite-sample effects could be explicitly removed from the algorithms. This is most easily incorporated in our approach by splitting a given collection of latent variables *x* into *M* disjoint subsets $x^{\left(i\right)}$. We reorder the vector indices in such a way that the first components correspond to $x^{\left(1\right)},x^{\left(2\right)}$, and so on. Hence, we obtain $x=\{x^{\left(1\right)},x^{\left(2\right)},\ldots ,x^{\left(M\right)}\}$. A structured mean-field approach is enforced by imposing a block matrix structure for the update matrix $A_{MF}=A_{\left(1\right)}⊕\cdots ⊕A_{\left(M\right)}$, where ⊕ is the direct sum operator. It is easy to see that this construction corresponds to a related block structure of the Γ matrix in Equation (Equation 3). This means that the subsets of the random vectors are modeled as independent. Hence, when the number of particles grows to infinity, one recovers the fixed-point equations for the optimal MF structured Gaussian variational approximation from our approach. As previously, as the number of particles grows to infinity, we recover the optimal MF Gaussian variational approximation. Note that using a structured MF does not change the complexity of the algorithm but requires fewer particles to obtain a full-rank solution.

### 3.6. Comparison with SVGD

Given the similarities with the SVGD methods [24], one could question the differences of our approach. The model proposed by [10] using a *linear kernel* $k\left(x,x^{\prime }\right)=x^{⊤}x^{\prime }+1$ has similar properties to our approach. The variable update becomes:$$\begin{array}{cc} \frac{dx}{dt} & =\frac{1}{N}\sum_{i=1}^{N}\left(-k\left(x_{i},x\right)\nabla \varphi \left(x_{i}\right)+\nabla_{x_{i}}K\left(x_{l},x_{i}\right)\right)\\ \; & =E_{\hat{q}}\left[I-\nabla \varphi \left(x\right)x^{⊤}\right]x-E_{\hat{q}}\left[\nabla \varphi (x)\right] \end{array}$$The fixed points are   
$$\begin{array}{cc} 0= & E_{\hat{q}}\left[\nabla \varphi (x)\right]\\ I= & E_{\hat{q}}\left[\nabla \varphi \left(x\right)x^{⊤}\right]=E_{\hat{q}}\left[\nabla \varphi \left(x\right){\left(x-m\right)}^{⊤}\right] \end{array}$$
where the last equality holds since $E_{\hat{q}}\left[\nabla \varphi (x)\right]=0$. This is the same as our algorithm fixed points (Equation 9). Similarly to Theorem 1, D+1 particles will converge to the exact *D*-dimensional multivariate Gaussian target. However, the generated flows are different. The main difference is that *we normalize our flow via the $L_{2}$ norm*, whereas [10] rely on the *reproducing kernel Hilbert space (RKHS) norm*, i.e., ${∥\varphi ∥}_{k}^{2}=\varphi^{⊤}K^{-1}\varphi$ where $\varphi_{i}=\varphi \left(x_{i}\right)$ and $K_{ij}=k\left(x_{i},x_{j}\right)$. For a full introduction on RKHS, we recommend [34]. Remarkably, centering the particles on the mean, namely, using the modified linear kernel $k\left(x,x^{\prime }\right)={\left(x-m\right)}^{⊤}\left(x^{\prime }-m\right)+1$, leads to the same dynamics. Additionally, when using SVGD, there is no direct possibility of computing the current KL divergence between the variational distribution and the target, unless some values are accumulated [35]. There is also no clear theory explaining what happens when the number of particles is smaller than the number of dimensions, for both distance-based kernels and the linear kernel.

## 4. Experiments

We now evaluate the efficiency of GPF and GF. First, given a Gaussian target, we compare the convergence of our approach with popular VGA methods, which are all described in Section 2. Second, we evaluate the effect of varying the number of particles for both Gaussian targets and non-Gaussian targets, especially with a low-rank covariance. Then, we evaluate the efficiency of our algorithm on a range of real-world binary classification problems through a Bayesian logistic regression model and a series of BNN on the MNIST dataset.

All the Julia [36] code and data used to reproduce the experiments are available at the Github repository: https://github.com/theogf/ParticleFlow_Exp (accessed on 27 July 2021).

### 4.1. Multivariate Gaussian Targets

We consider a 20-dimensional multivariate Gaussian target distribution. The mean is sampled from a normal Gaussian $\mu ∼N(0,I_{D})$ and the covariance is a dense matrix defined as $\Sigma =U\Lambda U^{⊤}$, where *U* is a unitary matrix and Λ is a diagonal matrix. Λ is constructed as $\log_{10}\left(\Lambda_{ii}\right)=\frac{\log_{10}\left(\kappa \right)\left(i-1\right)}{D-1}-1$ where κ is the condition number, i.e., $\kappa =\Lambda_{\max}/\Lambda_{\min}$. This means that, for κ=1, we obtain a Σ=0.1I, and for κ=100, we obtain eigenvalues ranging uniformly from 0.1 to 10 in log-space.

We compare GPF and GF to the state-of-the art methods for VGA described in Section 2, namely Doubly Stochastic VI **(DSVI)** [14], Factor Covariance Structure **(FCS)** [15] with rank p=D, iBayes Learning Rule **(IBLR)** [17] with a full-rank covariance and their Hessian approach, and Stein Variational Gradient Descent with both a linear kernel **(Linear SVGD)** [10] and a squared-exponential kernel **(Sq. Exp. SVGD)** [24]. For all methods, we set the number of particles or, alternatively, the number of samples used by the estimator, as D+1, and use standard gradient descent ($x^{t+1}=x^{t}+\eta \varphi^{t}\left(x^{t}\right)$) with a learning rate of η=0.01 for all particle methods. We use RMSProp [37] with a learning rate of 0.01 for all stochastic methods. We run each experiment 10 times with 30,000 iterations, and plot the average error on the mean and the covariance with one standard deviation. For GPF, we additionally evaluate the method with and without using natural gradients for the mean (i.e., pre-multiplying the averaged gradient with $C^{t}$), indicated, respectively, with a dashed and solid line. Figure 2 reports the $L_{2}$ norm of the difference between the mean and covariance with the true posterior over time for the target condition number κ∈{1,10,100}.

As Theorem 1 predicts, GPF converges exactly to the true distribution, regardless of the target. GF and other methods based on stochastic estimators cannot obtain the same precision as their accuracy is penalized by the gradient noise. IBLR approximate the covariance perfectly, despite the stochasticity of its estimator; however IBLR needs to compute the true Hessian at each step. When using a Hessian approximation instead, IBLR performed just like DSVI; the true benefit of IBLR appears when second-order functions are computed, which is naturally intractable in high-dimensions. SVGD with a linear kernel, achieves a good performance but is highly unstable: most of the runs (ignored here) diverge. This is due to the dot computation $x^{⊤}x$ which can become extremely high, especially for non-centered data. For this reason, we do not consider this method for the later experiments. SVGD with a sq. exp. kernel obtains a good estimate for the mean but fails to approximate the covariance.

Perhaps surprisingly, GF does not perform much better than DSVI or FCS. This is potentially due to *the benefit of Riemannian gradients being canceled by the gradient noise* [38] providing a strong argument for particle-based methods over stochastic estimators.

Remarkably, we also confirm Theorem 2, that the convergence speed of $C^{t}$ is independent of the target Σ, while the convergence speed of $m^{t}$ has this dependency unless the natural gradient is used (see the dashed curves). The case κ=1 highlights that *natural gradient do not necessarily improve convergence speed*.

### 4.2. Low-Rank Approximation for Full Gaussian Targets

We explore the effect of the number of particles for both Gaussian and non-Gaussian targets. We use the same Gaussian target from the previous experiment in 50 dimensions with a full-rank covariance determined by their condition number $\kappa =\frac{\lambda_{\max}}{\lambda_{\min}}$. The covariance eigenvalues $\lambda_{i}$ in log-space range uniformly from 0.1 to 0.1κ. For a given target multivariate Gaussian, we vary the number of particles from 2 to D+1 and look at the absolute difference of |tr(C−Σ)|. The results in D=50, as well as the corresponding predictions (in dashed-black), from Theorem 3, are shown on Figure 3.

The empirical results perfectly match the theoretical predictions, confirming that, for Gaussian targets, the particles determine a low-rank approximation whose spectrum is equal to the largest eigenvalues from the target.

### 4.3. High-Dimensional Low-Rank Gaussian Targets

We consider a typical low-rank target case where the dimensionality is high but the effective rank of the covariance is unknown. The target is given by p(x)=N(μ,Σ) where $\mu ∼N\left(0,I_{D}\right)$, the covariance is defined by $\Sigma =U\Lambda U^{⊤}$, where *U* is a D×D unitary matrix and Λ is a diagonal matrix defined by
$$\begin{array}{c} \Lambda_{ii}=\left\{\begin{array}{cc} N(2,1), & \mathrm{if}\;i\leq K\\ 10^{-8}, & \mathrm{otherwise} \end{array}\right. \end{array}$$
where *K* is the effective rank of the target. We pick D=500 and vary K∈{10,20,30} to simulate a true problem where the correct *K* is not known. We test all methods allowing for low-rank structure, namely, GPF, GF, FCS and SVGD (Linear and Sq. Exp.). We fix the rank (or the number of particles) to be 20; therefore, we obtain three cases where the rank is exact, under-estimated, and over-estimated. For all methods, we use RMSProp [37] for the stochastic methods, or a diagonal version of it (see Section 4.4) for the particle ones. The error of the mean and the covariance is shown in Figure 4. Note that the difference in the initial error on the covariance is due to the difficulty of starting with the same covariance between particle and stochastic methods.

We observe once again that the SVGD with a linear kernel fails to converge due to the large gradients. All methods perform equally in the estimation of the mean while being non-influenced by the rank of the target. As expected, the approximation quality for the covariance degrades when the rank gets bigger, but all algorithms still converge to good approximations. SVGD with a sq. exp. kernel performs much worse than the rest of the methods. This is a known phenomenon where, for high dimensions, the covariance SVGD is either over- or underestimated.

### 4.4. Non-Gaussian Target

We now investigate the behavior of our algorithm with non-Gaussian target distributions. We built a two-dimensional banana distribution: $p\left(x\right)\propto \exp(-0.5\left(0.01x_{1}^{2}+0.1{\left(x_{2}+0.1x_{1}^{2}-10\right)}^{2}\right))$, varied the number of particles used for GPF in {3,5,10,20,50} and compared it with a standard full-rank VGA approach. We also showed the impact of replacing a fixed η with the Adam [39] optimizer for 50 particles. The results are shown in Figure 5. As expected, increasing the number of particles madesthe distribution obtained via GPF increasingly closer to the optimal standard VGA, even in a non-Gaussian setting. However, using a momentum-based optimizer such as Adam breaks the linearity assumption of the original flow (Equation 5) and leads to a twisted representation of the particles. (We observed the same behavior with other momentum-based optimizers). A simple modification of the most known optimizers allows the linearity to be maintained while correctly adapting the learning rate to the shape of the problem. Most optimisers accumulate momentum or gradients element-wise, and end up modifying the updates as $x^{t+1}=x^{t}+P^{t}⊙\varphi^{t}\left(x^{t}\right)$, where $P^{t}\in R^{D\times D}$ is the preconditioner obtained via the optimiser and ⊙ is the Hadamard product. By instead taking the average over each dimensions, we obtained the updates $x^{t+1}=x^{t}+P^{t}\varphi^{t}\left(x^{t}\right)$, where $P^{t}$ is a D×D diagonal matrix. The details of the dimension-wise conditioners for ADAM, AdaGrad and AdaDelta are given in Appendix H.

### 4.5. Bayesian Logistic Regression

Finally, we considered a range of real-world binary classification problems modeled with a Bayesian logistic regression. Given some data ${\left\{\left(x_{i},y_{i}\right)\right\}}_{i=1}^{N}$ where $x_{i}\in R^{D}$ and y∈{−1,1}, we defined the model $y_{i}∼\mathrm{Bernoulli}\left(\sigma \left(w^{⊤}x_{i}\right)\right)$ with weight $w\in R^{D}$, and with σ being the logistic function. We set a prior on *w*: $w\;N\left(0,10I_{D}\right)$. We benchmarked the competing approaches over four datasets from the UCI repository [40]: spam (N=4601,D=104), krkp (N=351,D=111), ionosphere (N=3196,D=37) and mushroom (N=8124,D=95). We ran all algorithms discussed in Section 4.1, both with and without a mean-field approximation; SVGD was omitted since it is too unstable. All algorithms were run with a fixed learning rate $\eta =10^{-4}$, and we used mini-batches of size 100. We show alternative training settings in Appendix I. Note that FCS, for mean-field, simplifies to DSVI Additionally, we did not consider full-rank IBLR, as it is too expensive, and we used their reparametrized gradient version for the Hessian. Figure 6 shows the average negative log-likelihood on 10-fold cross-validation with one standard deviation for each dataset. While, as expected, the advantages shown for Gaussian targets do not transfer to non-Gaussian targets, GPF and GF are consistently on par with competitors. On the other hand, IBLR tends to be outperformed. It is also interesting to note that mean-field does not seem to have a negative impact on these problems, and performance remains the same even with a full-rank matrix.

### 4.6. Bayesian Neural Network

We ran our algorithm on a standard network with two hidden layers each, with L=200 neurons and tanh activation functions (we additionally tried ReLU [41], but some baselines failed to converge). We trained on the MNIST dataset [42] (N= 60,000, D=784) and used an isotropic prior on the weights $p\left(w\right)=N\left(0,\alpha I_{D}\right)$ with α=1.0. We additionally compared these with *Stochastic Weight Averaging-Gaussian*
**(SWAG)** [27] with an SGD learning rate of $10^{-6}$ (selected empirically) and *Efficient Low-Rank Gaussian Variational Inference*
**(ELRGVI)** [26]. We varied the assumptions on the covariance matrix to be diagonal **(Mean-Field)**, or to have rank L∈{5,10}. Additionally, we showed, for GPF, the effect of using a structured mean-field assumption by imposing the independence of the weights between each layer (**GPF (Layers)**).

We trained each algorithm for 5000 iterations with a batchsize of 128(∼10 epochs) and reported the final average negative log-likelihood, accuracy and expected calibration error [43] on the test set (N= 10,000) on Table 1. The predictive distribution is given by
$$\begin{array}{c} p\left(y=k|x^{*},D\right)=\int p\left(y=k|x^{*},w\right)p\left(w|D\right)dw\approx \int p\left(y=k|x^{*},w\right)q\left(w\right)dw, \end{array}$$
where D is the training data, and $x^{*}$ is a test sample. We computed the accuracy and the average negative test log-likelihood as:$$\begin{array}{cc} \mathrm{Acc} & =\frac{1}{N}\sum_{i=1}^{N}1_{y_{i}}\left(\arg_{k}\max p\left(y=k|x_{i}^{*},D\right)\right)\\ \mathrm{NLL} & =-\frac{1}{N}\sum_{i=1}^{N}\log p\left(y=y_{i}|x_{i}^{*},D\right) \end{array}$$
where $1_{y}\left(x\right)$ is the indicator function (equal to 1 for y=x, 0 otherwise). For the definition of expected calibrated error, we refer the reader to [43]. Additional convergence and uncertainty calibration plots can be found in Appendix I.

Overall, the SVGD method performed best in terms of both accuracy and negative log-likelihood. However, SVGD is not in the same category as others, since it is not a VGA. For VGAs, we observed that a low-rank approximation improves upon mean-field methods. In particular, assuming independence between layers provides a large advantage to GPF. GPF and GF generally perform equally or better than all the other VGA methods. Note that, although not reported here, all methods needed approximately the same time for the 5000 iterations, except for SWAG, which only needed the MAP and a few thousand iterations of SGD afterward, making it generally faster but also less controlled (a grid search was needed to find the appropriate learning for SGD).

## 5. Discussion

We introduced GPF, a general-purpose and theoretically grounded, particle-based approach, to perform inference with variational Gaussians as well as GF its parameter version. We were able to show the convergence of the particle algorithm based on an empirical approximation of the free energy. We also showed that we can approximate high-dimensional targets by allowing for low-rank approximations with a small number of particles. The results for Gaussian targets suggest that the convergence of posterior covariance approximation may relax asymptotically fast, with small dependence on the target. This work is the first step in analyzing convergence speed and guarantees in inference with variational Gaussians, and future work could extend guarantees to non-Gaussian problems. One could also take advantage of existing particle-based VI methods to accelerate inference further or reach a better optima [44,45].

## Figures and Tables

**Figure 1 entropy-23-00990-f001:**
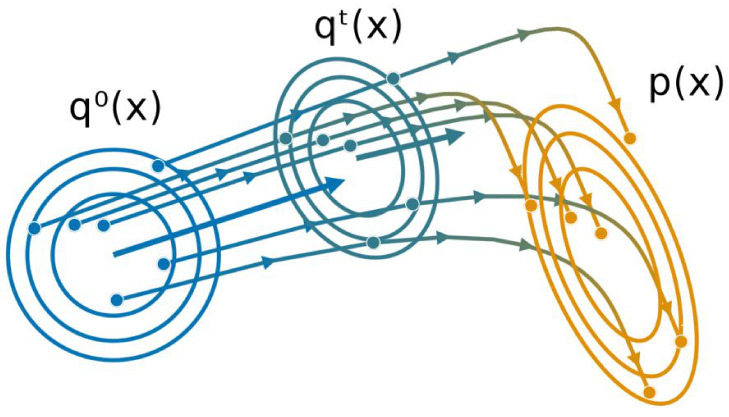
Illustration of the Gaussian Particle Flow algorithm, with $q^{0}\left(x\right)$ and p(x) representing the initial and target distribution respectively. Particles are iteratively moved according to the gradient flow starting from $q^{0}\left(x\right)$, approximating a new Gaussian distribution $q^{t}\left(x\right)$ at each iteration *t*.

**Figure 2 entropy-23-00990-f002:**
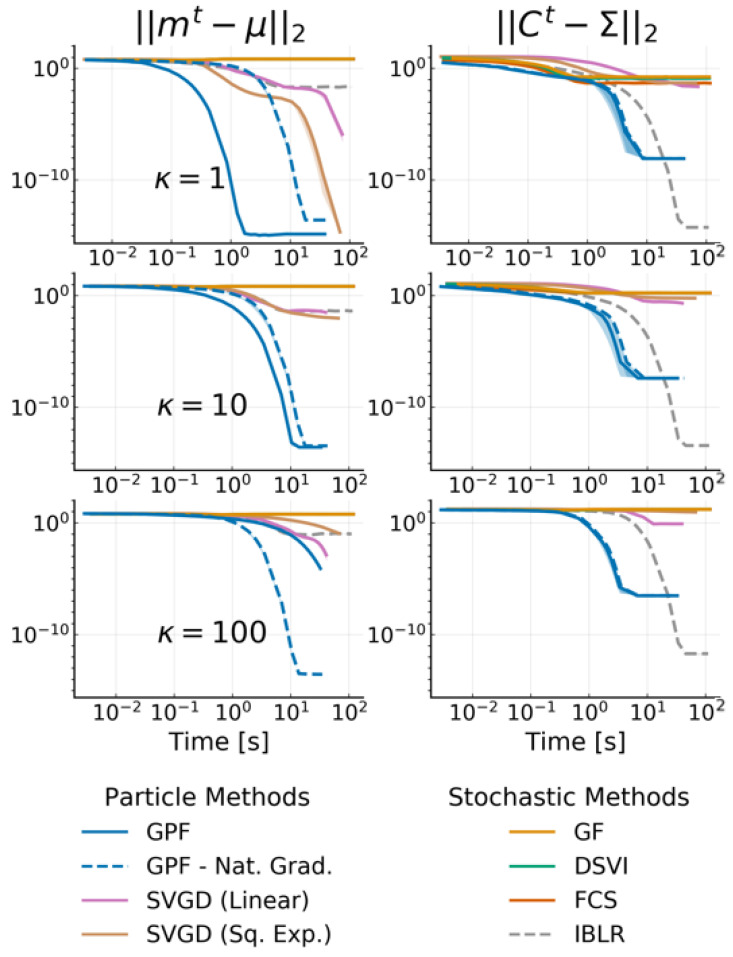
$L^{2}$ norm of the difference between the target mean μ (left side) and target covariance Σ (right side) with the inferred variational parameters $m^{t}$ and $C^{t}$ against time for 20-dimensional Gaussian targets with condition number κ. We use D+1 particles/samples and show the mean over 10 runs as well as the 68% credible interval. Methods with dashed curves use natural gradients on the mean. Note that DSVI, GF and FCS are overlapping and are, at this scale, indistinguishable from one another.

**Figure 3 entropy-23-00990-f003:**
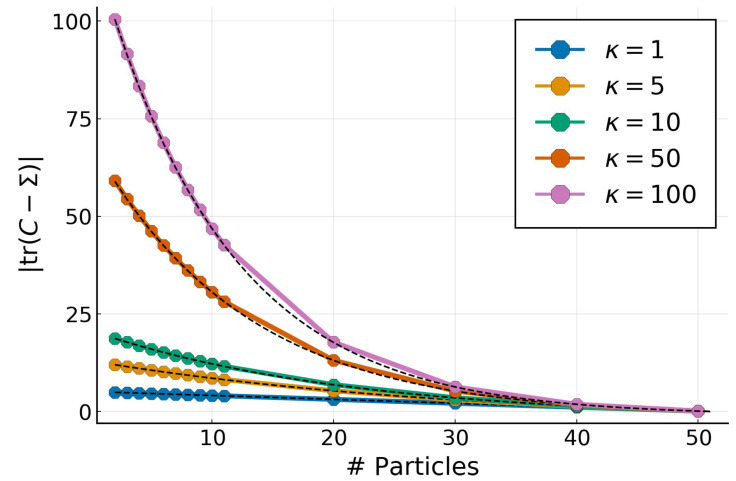
Trace error for a Gaussian target with D=50 and condition numbers κ for a varying number of particles with GPF. Predictions from Theorem 3 are shown in dashed-black.

**Figure 4 entropy-23-00990-f004:**
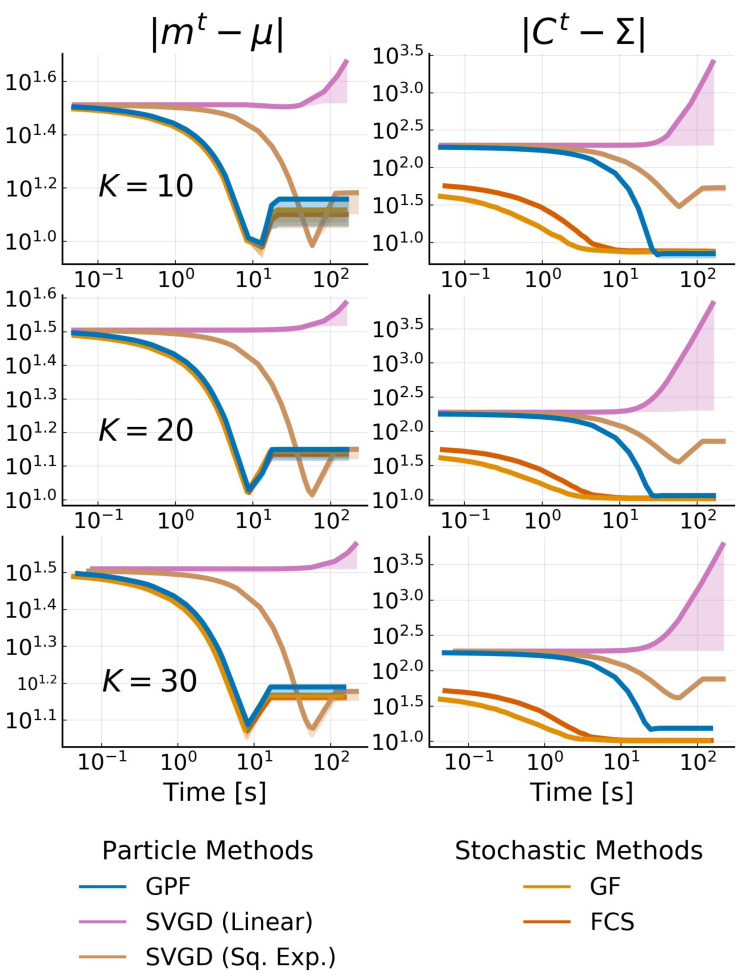
Convergence plot of low-rank methods for a 500-dimensional multivariate Gaussian target with effective rank K∈{10,20,30}. The rank of each method is fixed as 20. The difference in the starting point for the covariance is due to the initialization difference between each method. We show the mean over 10 runs for each method with shadowed areas representing the 68% credible interval.

**Figure 5 entropy-23-00990-f005:**
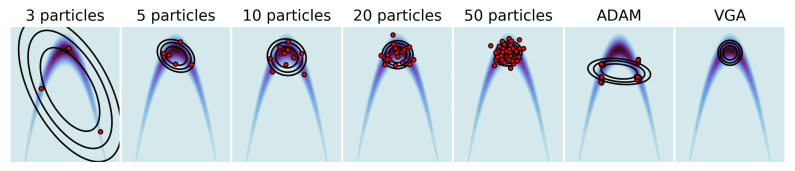
Two-dimensional Banana distribution. Comparison of GPF using an increasing number of particles and a different optimizer (ADAM) with the standard VGA (rightmost plot).

**Figure 6 entropy-23-00990-f006:**
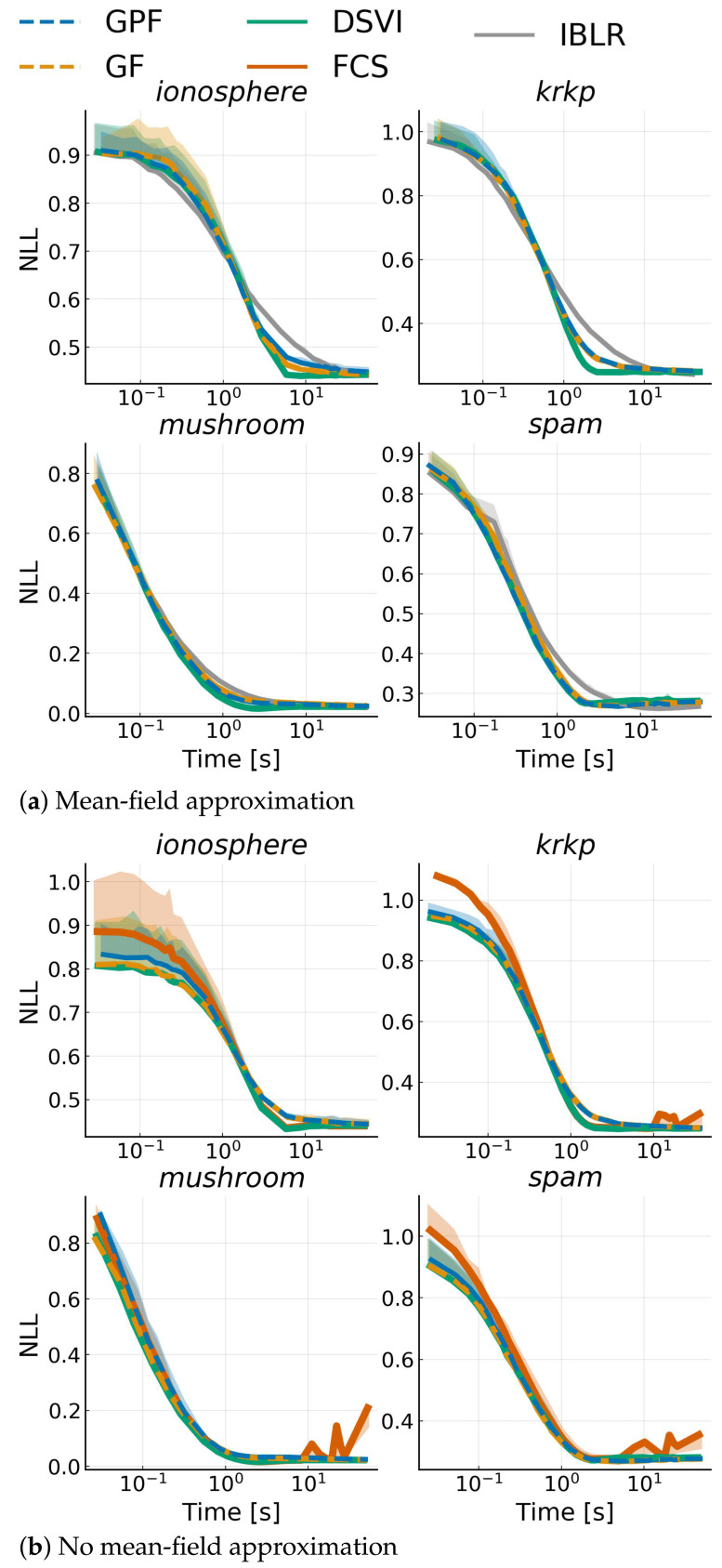
Average negative log-likelihood vs. time on a test-set over 10 runs against training time for a Bayesian logistic regression model applied to different datasets. Top plots use a mean-field approximation, while bottom plots use a low-rank structure for the covariance with rank L=100.

**Table 1 entropy-23-00990-t001:** Negative Log-Likelihood (NLL), Accuracy (Acc), and Expected Calibration Error (ECE) for a Bayesian Neural Networks **(BNN)** on the MNIST dataset. We varied the rank of the variational covariance from mean-field (all variables are independent) to a low-rank structure with L∈{5,10}. Bold numbers indicated the best performance, and italic bold numbers indicate the best performance when restricted to VGA methods. Convergence and calibration plots can be found in Appendix I.

Alg.	Mean-Field			L=5			L=10		
	NLL	Acc	ECE	NLL	Acc	ECE	NLL	Acc	ECE
GPF	0.183	0.95	0.0384	0.166	***0.96***	0.0918	0.172	0.955	0.0869
GPF (Layers)	-	-	-	***0.147***	0.958	**0.0181**	0.178	0.952	0.0395
GF	0.178	0.953	0.0706	0.185	0.956	0.136	0.171	0.952	0.0455
DSVI	0.204	0.945	0.11	-	-	-	-	-	-
SVGD (Sq. Exp)	-	-	-	0.139	0.965	0.0732	**0.133**	**0.967**	0.0879
SWAG	-	-	-	0.257	0.957	0.0662	0.287	0.956	0.0878
ELRGVI	-	-	-	0.453	0.901	0.53	0.537	0.882	0.777

## Data Availability

Datasets can be found on the UCI dataset website [40] and the MNIST dataset can be found on Yann Lecun website [42].

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
