# Peer review of "Flexible and Efficient Inference with Particles for the Variational Gaussian Approximation"

_entropy, 2021, doi:10.3390/e23080990_

Round 1

Reviewer 1 Report

In this paper the authors consider a Gaussian Particle Flow (GPF) framework for approximating a Gaussian variational distribution with particles. The theoretical properties of the proposed approach are studied, and the method is compared with other existing approaches using simulated and real-world data.   

I think the paper is well-written and suitable for publishing in the special issue. I only have some questions and comments that should be addressed before publishing the paper:

1. To attract reader from statistics community, please use proper statistical terminology throughout the paper and check the notations carefully. For example, it is sometimes difficult to follow the mathematical notations as dimensions of random vectors and matrices are not given. 

2. Unify the referring style. Sometimes you just refer by using the numbers of articles, like [1], and sometimes using also names, like Challis and Barber [13]. 

3. I have some problems in understanding your simulation results. When referring to Figure 2 you mention that you plot "the average error on the mean and the covariance with 1 standard
deviation", but in the figure legend you mention RMSE? What is exactly plotted on the y-axis? Why 1 standard deviation? Some stochastic methods are not visible in the plots. Perhaps you could use not so bolds lines to make these visible? And lastly, why only ten replications? I think this is not acceptable for valid inference. I would have expected (at least) hundreds.   

4. Related to previous comment, how are the accuracy and expected calibration error computed in Section 4.6? 

5. Please check the paper very carefully for language. 

Minor comments: 

1. p1, l1: What do you mean by "well-calibrated" here? Please use proper statistical terminology.  

2. p1, l33: Should this be "..do not have this issue and are proved to be"?

3. p2, l42: Should this be "mean-field approximations"?

4. p2, l53: prior term -> prior distribution

5. p3, l51: Please explain what does the ELBO stand for.

6. p3, l58: Please explain using proper statistical language what do you mean by "are too expensive in a black box setting".

7. p3, l66 (and elsewhere): Is it clear that you can change the order of integration and differentiation here?

8. p3, l69: Where does the dimension P come from? Should you have C = \Gamma\Gamma' otherwise the dimensions (with diag(d)) do not match?

9. p6: Why do you use two different notations for taking the derivative, d/dt and \partial/\partial t,  (and also \nabla)?

10. p9, eq (16): I do not understand, where this comes from. Please explain.

11. p9, line 183: Is the covariance matrix for normal distribution the identity matrix or something else? If the identity, then several notations for it are used in the paper - with and without the dimension. 

12. p10, Thm 2: What do you mean by notation N(x|\mu,\Lambda^{-1})? Please unify notations for Gaussian distributions. 

13. p12, l259: What is the RKHS norm? 

14. p13, l283-286: Are all these methods described in Section 2?

15. p15, l331: Give \Lambda_{ii} as one-line formula.

16. p16, Figure 4: What is the shadowed area in the plot? Please make sure that the figure legend is self-standing. 

17. p17, Figure 5: Give the figure later in Section 4.4. 

Author Response

Thank you very much for your swift and detailed review. In what follows, we address your comments in detail.

  •  "please use proper statistical terminology": We corrected inconsistencies and inaccuracies, and have incorporated your comments in the paper.
  • "Unify the referring style": We corrected the style of citations.
  • About the results in Figure 2: we are plotting the square root of the L2 norm between the inferred parameter and the true parameters. We will emphasize this both in the caption and the axis. We would also like to clarify that we are not plotting the standard deviation but the 68% credible interval, which we will reflect in the text. We will also improve its visibility, especially for DSVI. The reason for 10 replications is that the variance across runs is already small and we did not observe benefits from running additional simulations.
  • In section 4.6, the accuracy was computed as "number of correctly predicted samples / total number of samples", the NLL is the average negative loglikelihood, i.e. $p(y=y_{true}|\theta)$, for the ECE we refer to the appropriate litterature. We will write this more explicitly in the text.
  • "Is it clear that you can change the order of integration and differentiation here?" Changing the order of integration and differentiation is not the issue here since the bounds of the integral do not depend on the variational parameters. The issue rather comes from the gradients of $q$ that would appear. Reparametrization allows us to avoid that.
  • We also thank you for the rest of your more detailed comments, which we incorporated by improving the text and figures.

Reviewer 2 Report

I found the paper to be interesting, well written and well executed.  The paper synthesizes variational inference (VI) that approximate a distribution, usually the posterior distribution p(x|y), with particle methods.  The underlying idea for VI is that the target distribution is too difficult or too expensive to compute.   It replaces the difficult, target distribution with a simpler one that is as close as possible to the target where “close” is measured by entropy or Kullback-Liebler (KL) divergence. 

Particle methods approximate the target distribution with a set of numbers, hopefully a random sample from the distribution.  Particle filter algorithms originated for Bayesian analysis of nonlinear dynamic models. A set of particles are generated at time 0.  These particles are updated over time as the system evolves.   At any given time, they are then used in importance sampling to approximate integrals with respect to the target distribution.

In this paper, an initial set of particles are generated.   New particle are derived from the old ones by using the gradient of phi(x) = - log(p(y|x)p(x)) where the target distribution is p(x|y) = p(y|x)p(x)/p(y).  The updating equations are similar to numerical methods for finding MLE or MAP but instead of finding the optimizing values, the output is either a random sample from the target distribution (Algorithm 1) or the VI Gaussian approximation (Algorithm 2).  The paper provides empirical studies that demonstrate the proposed algorithms perform well when compared to existing methods.

Comments:

  1. The algorithms do not include stopping rules: T is a parameter for the algorithm.  Is it possible to recommend smart stopping rules?
  2. Would it be useful to incorporate ideas from particle filtering? For instance, it may be possible to use a smaller N or T by computing importance weights and approximating integrals with importance sampling approximations instead of running the algorithms all the way to obtaining random samples.
  3. Would periodically adding a Metropolis step (say, every 10th iteration) for a subset of particles in GPF improve the algorithm’s performance?
  4. Are there conditions where the methods fail? For instance, are there circumstances where the particles become degenerate, or some particles trundle off to infinity?
  5. Page 8. The paper recommends replacing zero eigenvalues of the empirical covariance matrix in Equation (15) by 1. This approach seems arbitrary.   It increases the variance of the Gaussian distribution, but the amount of increase seems arbitrary.  For instance, if the trace of the covariance is 0.01, then the new covariance is much larger.  Alternatives would reallocate factor*(trace of covariance) to the zero and non-zero eigenvalues where 1 < factor < 1.5 (say).   
  6. The empirical covariance matrix can become singular because 1) there are not enough particles, 2) the particles become degenerate because the algorithm fails, 3) the likelihood function is poorly specified and the model is not identified or 4) Gaussian VI is a poor approximation. 1) is simple to handle by increasing N. There are various methods to remedy 2), such as stochastically modifying some of the particles.  (Metropolis step?).    3) is more problematic, especially since many ML models are not identified.  4) is very tough.  So, a singular covariance matrix may be diagnostic about the model or algorithm.  Any insights?

Author Response

Thank you very much for your swift and detailed review. In what follows, we address your comments in detail.

  • You are correct that particles are updated at every step. However, importance sampling is not used to estimate integrals (although it could be done). Instead, we are directly using the particles as an approximation to the true target distribution. Additionally, the dynamics in algorithms 1 and 2 differ from MLE or MAP in the sense that we want to minimize the KL divergence between our variational distribution and the target distribution. Algorithm 1 provides an empirical approximation to a Gaussian variational distribution and is not guaranteed to correspond to samples from the target distribution.
  • "Is it possible to recommend smart stopping rules?": This is a good point. One can easily compute the variational free energy $F^t$ at every step and check for an easy stopping criteria like $|F^{t+1} - F^{t}| < \epsilon$.
  • "Would it be useful to incorporate ideas from particle filtering?": We thank the reviewer for the interesting connection. One would need to address how the Gaussianity of the distribution would be maintained by using different weights for each particles. An additional challenge is that using a smaller $N$ would lead to lower-rank covariance.
  • "Would periodically adding a Metropolis step (say, every 10th iteration) for a subset of particles in GPF improve the algorithm’s performance?": At any time step all particles participate equally in approximating the variational distribution. Therefore, it is not clear how a Metropolis step could be used and how effective it would be. We considered how the initial samples could eventually be taken and leave this as an interesting avenue for future work.
  • "Are there conditions where the methods fail?": Yes. Since we are appoximating continuous dynamics by an Euler discretization, wrong stepsizes might lead to divergences (like particles going to infinity). However, using the optimizers proposed and an appropriate step size solves this problem. Another potential issue is when the target is highly "complex" and a Gaussian distribution is too unfit to approximate it. But this issue is common to all GVA.
  • "The paper recommends replacing zero eigenvalues of the empirical covariance matrix in Equation (15) by 1": This replacement is only happening when computing a *surrogate* function of the variational free energy. When making predictions, we are using the *exact* empirical covariance.
  • "The empirical covariance matrix can become singular": The covariance matrix is guaranteed to be singular if $N < D + 1$. However, this is not a problem, since it only means that the covariance matrix lies in a low-rank subspace of $R^D$. The problem of particles becoming degenerate is not a concern in practice, and it would be caused by a highly ill-posed target distribution. While non-Gaussianity does not necessarily mean that the problem is ill-posed, your points 3) and 4) could be merged into one analysis tool where we could assess the Gaussianity of the target distribution.

Round 2

Reviewer 2 Report

Sorry for the misunderstanding.  My comment on importance sampling related to the empirical approximation q where the support points are {xi}.  From my perspective as a Bayesian statistician, the major use of q is approximating integrals:  E[H(x)|Data] ~ N^{-1} sum_i H(xi) by integrating over the empirical distribution.   In the simplest case H(x) = I(x in A) indicated set membership and P(X in A | Data) = E[H(x)|Data].     

My thought was that there may be an opportunity to improve the approximation with sum_i H(xi)*wi/ sum_i wi where wi are the importance sampling weights.  Of course, if {xi} is a random sample from p, the then wi = 1.  

I appreciate that the goal of the paper is approximating the target distribution and not integrals, but if one takes the viewpoint of Bruno DeFinetti, distributions are merely intermediate devices to  compute posterior expectations.  

Anyway, I enjoy your research and wish you the best of luck.  I hope the recent flooding In Germany did not effect your friends and family.